# Determinant Factors of Voluntary or Mandatory Vaccination against COVID-19: A Survey Study among Students at Albanian University

**DOI:** 10.3390/vaccines11071215

**Published:** 2023-07-07

**Authors:** Elona Kongo, Kleva Shpati, Aida Dama, Sibela Ymeraj, Ema Murati, Uels Veliaj, Sonela Xinxo

**Affiliations:** 1Dentistry Department, Faculty of Medical Science, Albanian University, 1017 Tirana, Albania; s.xinxo@albanianuniversity.edu.al; 2Pharmacy Department, Faculty of Medical Science, Albanian University, 1017 Tirana, Albania; k.shpati@albanianuniversity.edu.al (K.S.); a.dama@albanianuniversity.edu.al (A.D.); 3Department of Psychology, Faculty of Social Science, Albanian University, 1017 Tirana, Albania; s.ymeraj@albanianuniversity.edu.al (S.Y.); e.murati@albanianuniversity.edu.al (E.M.); 4IT Department, Albanian University, 1017 Tirana, Albania; uels.veliaj@albanianuniversity.edu.al

**Keywords:** COVID-19, vaccine, medical science students, mandatory policy

## Abstract

Introduction: The world faced serious health and socioeconomic issues with the advent of COVID-19. Along with handwashing, social distancing, and the wearing of masks, vaccination was considered by medical authorities as a key way to curb the spread of the virus. One of the measures that have been proposed to increase vaccinations is the mandatory policy. The purpose of this study was to compare the determinants of voluntary and mandatory vaccination among students at Albanian University. Methodology: This cross-sectional study, conducted by means of an online survey, was conducted among students enrolled at the private Albanian University, Tirana, Albania during the last week of the winter semester, 7–14 February 2022. Results: In sum, 878 students participated in this study, among whom 612 (69.7%) were females and 266 (30%) were males. Of the participants, 506 (57%) were enrolled in medical science (MS) and 372 (42.3%) were in the non-medical science (Non-MS) group. A total of 773 (88%) were vaccinated against COVID-19, and 105 (11.8%) were not vaccinated. A total of 466 (53%) reported voluntary vaccination, and 412 (46.9%) reported mandatory vaccination. Among the students who were vaccinated voluntarily, 266 (57%) were from the MS group and 200 (42.9%) were from the Non-MS group. A total of 237 (57.5%) of students in the mandatory vaccination group were from the MS group, and 175 (42.4%) were from the Non-MS group. Conclusions: Vaccine safety and efficacy were hindering factors in vaccination. Additionally, based on the results of this study, the students felt encouraged by the academic staff to vaccinate. This clearly demonstrates that the staff does not lack the skills to enhance students’ knowledge about the risk of infectious diseases and the importance of vaccination. Therefore, to influence students’ attitudes as much as possible toward vaccination, comprehensive educational programs, including modifications of existing curricula, should be considered.

## 1. Introduction

The world faced serious health and socioeconomic issues with the advent of COVID-19. Along with handwashing, social distancing, and the wearing of masks, vaccination was considered by medical authorities as a key way to curb the spread of the virus. Efforts to enable the development of vaccines were documented by the WHO: on 26 April 2020, there were 52 possible vaccine candidates, with 7 in the clinical evaluation phase [1]. Regardless of the crucial role vaccination plays in preventing infectious diseases, vaccine hesitancy (VH), which “refers to delay in acceptance or refusal of vaccination despite availability of vaccination services,” was recognized by the WHO as “one of the top-ten threats for global health” [2]. Hence, although a vaccine was not available, researchers worldwide started evaluating the rate of VH and VA (vaccine acceptance) for a possible COVID-19 vaccine [3,4,5]. Almost in accordance, in addition to evaluating different target groups, safety-related fear, lack of adequate information, and conspiracy were acknowledged as vaccination-hindering factors [6,7,8].

For many reasons related to disease severity and the crucial role in preventing the spread of infection [9] and previous vaccine hesitancy [10,11], many studies focused on students. The overall rate of COVID-19 VH among 31,948 college/university students around the world was 22%, with contributing factors such as safety-related fear and misinformation [12].

Even the field of study, especially medicine-related, was taken into consideration as a factor in vaccination attitudes [13,14,15]. Bearing in mind that those students are future medical personnel, whose attitudes may impact people’s vaccination choices [7], the study of this target group becomes even more important. In this context, the reported data on vaccination among healthcare students seem to provide optimistic results [16,17]. Regardless, higher rates of VH were reported among medicine-related students. The 22.5% VH rate among dental students worldwide was related to socioeconomic factors, social media, and insufficient knowledge [18]. In another review, a 25.8% VH rate was reported among healthcare students concerned about side effects [19].

The recognition of vaccine hesitancy posed a barrier to the successful implementation of vaccination; hence, a mandatory policy was included among measures to increase vaccination. Mills [20] suggested that mandatory certification could be one mechanism to increase uptake among younger people and certain groups, such as men and those from low socioeconomic backgrounds, to reach population-level immunity and to protect the broader population. Graeber [21] stated that a mandatory vaccination would almost certainly achieve herd immunity against COVID-19, since all those for whom there is no medical contraindication would also get vaccinated.

On 9 March 2020, the day of the confirmed first case of infection with SARS-CoV-2 in Albania, universities and other educational institutions were initially closed for two weeks, but this restrictive measure was extended such that the second semester of the 2019–2020 academic year was conducted by means of online lectures. Regarding the academic year 2020–2021, health authorities considered that the reopening of the universities would increase COVID-19 transmission, so the whole year was conducted online. Vaccination in Albania began in December 2020 with a target group of healthcare workers. In August 2021, 23.6% of the population had received at least one dose of the COVID-19 vaccine [22]. Hence, mandatory vaccination was announced for all healthcare workers, teachers, and students aged over 18, and facilities were adopted to better assist students [23]. The announcement regarding the mandatory policy was not supported by the students [23].

There is a lack of information about Albanian students’ attitudes toward COVID-19 vaccination. In a study performed by Patelarou [24] assessing nursing students from seven countries, 32.6% of 313 Albanian students responded positively to vaccination, which was considered among the lowest. Understanding factors that contribute to vaccine acceptance and hesitancy could help improve vaccination campaigns. The purpose of this study was to compare the determinants of voluntary and mandatory vaccination among students at Albanian University.

## 2. Materials and Methods

This cross-sectional study, utilizing a convenience sample, was conducted among students at Albanian University (AU), Tirana, Albania during the last week of the winter semester, from 7 to 14 February 2022. AU is a private university with 3849 students (2471 females and 1378 males) attending three faculties: the Faculty of Medical Science (MS), which includes dentistry, pharmacy, nursery, and dental technician programs, with 2150 students (1568 females and 582 males); the Faculty of Applied Science and Economics (ASE), with 1092 students (433 females and 659 males); and the Faculty of Social Science (SS), with 607 students (470 females and 137 males).

The questionnaire was developed according to WHO [25] guidelines and after an extensive literature review [13,14,26,27].

It consisted of 25 closed-ended questions addressing: (1) demographics (gender, age, study program); (2) COVID-19-related health experiences, fear at the beginning of pandemics, fear for one’s life if infected, observance of protective measures, being infected with COVID-19, severity of illness experienced in the family, and major sources of information; and (3) attitudes toward vaccination. They were also asked whether they had been vaccinated voluntarily, because of the mandatory policy, or influence from classmates, and about intentions to recommend vaccination to others, encouragement from academic staff, or hesitation over an unsafe and ineffective vaccine. No particular vaccine name was included.

A Microsoft Form link containing the questionnaire was emailed to a total of 3849 students enrolled in the academic year 2021–2022. Being enrolled in any study program offered by AU was the only inclusion criterion for the study. The email containing the link also contained information regarding the study, and students were asked to fill out the form only once. The form was designated to accept answers from 7 to 14 February 2022, the last week of the winter semester. A postponement was also foreseen if there were not enough answers. During the week the form was active, three email reminders were sent. By the end of the week, 934 returned forms were considered optimal, so there was no need to postpone acceptance.

The statistical analyses were conducted using SPSS 26. Means and standard deviations were used for continuous variables and frequencies for categorical variables. The statistical method employed uses hypothesis testing for the equality of proportion using the standard values method. The level of significance was set at an alpha value of 5%.

### Ethical Issues

The study was conducted in accordance with the ethical standards of the Declaration of Helsinki and approved by the Ethics Committee of Albanian University, ref. 66, 31 January 2022.

Participation was voluntary, and the return of the questionnaire was accepted as a form of individual consent to participate in the survey. Students were informed of the possibility of withdrawing, with no other consequences on their status or grades. No incentives were offered for taking part in this study.

## 3. Results

From the 934 questionnaires received (56 were not considered eligible: 45 had incomplete answers and 11 did not report sex), 878 were included in the study. Among them, 612 (68.8%) were from female students and 266 (29.2%) from male students. The average age 22.76 years, with a minimum of 18 years and a maximum of 29 years, and a standard deviation of 5.336 years. The majority (72.89%) were 18- to 23-year-olds. Overall, 506 (57.6%) of the participants were enrolled in MS (medical science students enrolled in one of the study programs of the Faculty of Medical Science), and 372 (42.3%) were non-MS (non-medical science students enrolled in one of the study programs of the Faculty of Applied and Economic Science and Social Science).

COVID-19-related health experiences are shown in Table 1. A total of 347 (37.8%) students had been infected in the last 6 months. A total of 298 (33.3%) stated that they had experienced SARS-CoV-2 disease in the family.

With regard to attitudes toward vaccination, as shown in Table 1, 851 (97%) had regular vaccinations in childhood, 773 (86.8%) were vaccinated against COVID-19, and 105 (11.8%) were not vaccinated. Based on the definition of VH, the sample was divided into two groups: those that declared mandatory vaccination—VH group 412 (46.9%); and those vaccinated voluntarily—VA group 466 (53%). To compare attitudes toward vaccination according to the study field, students in the VH and VA groups were divided into MS and Non-MS subgroups.

In Table 2, we present results obtained after comparison of determinants of VA according to the field of study (MS and Non-MS 466): 53% reported voluntary vaccination. A total of 302 (64.8%) were females, and 266 (57%) were from the MS group. A total of 207 (77.8%) MS students versus 146 (73%) from the Non-MS group declared experiencing fear at the beginning of pandemics. With regard to the observance of protective measures, positive answers were almost at the same level: 242 (90.9%) and 183 (91.5%), respectively. The questionnaire also asked about fearing for one’s life if infected and disease severity among family members. To the question of unvaccinated persons posing a risk to others, 145 (54.5%) from the MS group and 112 (56%) from the Non-MS group said yes. When asked about the influence of friends or classmates on their vaccination decision, 240 (90.2%) MS and 178 (89%) Non-MS students stated that there was no perceived influence. More than half of the students from both groups expressed their intention to recommend vaccination to friends or family members. A slight but not significant difference was observed with regard to the non-vaccination of family members (*p* = 0.3628). Interestingly, almost the same results were obtained regarding social networks as a major source of information: 201 (75.6%) MS and 149 (74.5%) Non-MS. Similar results, with no significant difference, were observed with regard to unsafe and ineffective vaccines (*p* = 0.395). The only significant change (*p* = 0.034) between the groups was with regard to encouragement from the academic staff.

Results obtained after comparing groups for mandatory vaccination are shown in Table 3. From the 878 students that participated in this study, 412 (46.9%) declared mandatory vaccination. A total of 286 (69.4%) were females and 126 (30%) were males. According to the study field, 237 (57.5%) were MS students and 175 (42.4%) were Non-MS students. A significant change was observed regarding fear experienced at the beginning of the pandemic. A total of 159 (67%) MS and 102 (58.2%) Non-MS students answered yes. With regard to the observance of protective measures, MS and Non-MS students had similar attitudes, reporting high levels of 199 (83.9%) and 151 (86.2%), respectively. MS students perceived more risk in everyday life (*p* = 0.002). A significant change (*p* = 0.0204) was observed in fear for one’s life if infected by COVID-19. More than half of the Non-MS group (99 (56.5%)) answered negatively, while 129 (54.4%) from the MS group answered positively. A total of 78 (33.4%) MS and 54 (30.8) Non-MS students had relatives severely affected by COVID-19. From the MS group, 47 (19.8%) reported a slight influence from classmates versus 21 (12%) from the Non-MS group, but the difference was not significant (*p* = 0.7338). To the question of whether the decision to vaccinate would have been easier if not mandatory, no significant change was observed (*p* = 0.7338), as 212 (89.5%) MS and 156 (89.2%) Non-MS students answered yes. Similarly, there was no significant change regarding social networks as a major source of information (*p* = 0.3576). From the results obtained, it is obvious that students from both groups are equally hesitant over an unsafe and ineffective vaccine (*p* = 0.6892). A significant change was observed with regard to the academic staff’s encouragement (*p* = 0.002), as 173 (73%) students from the MS group answered yes.

## 4. Discussion

This survey study conducted among students at Albanian University aimed to explore the factors related to mandatory and voluntary COVID-19 vaccination. In addition to the mandatory policy, 53% of participants in this study reported voluntary vaccination. In the majority of studies performed before the start of vaccination campaigns worldwide and the wide availability of vaccines, students showed VA of 52.8%, 58%, 60%, 69.3%, 73.3%, 80%, and 91% [5,13,14,15,28,29]. At the time this study was performed (February 2022), in Albania, 44.3% of the population was vaccinated [22]. Regarding students from AU, 1095 (28.4%) were vaccinated. In a previous study among Bulgarian students, 61.8% of students were vaccinated, while the country’s vaccination rate was 30.2%. According to the author, the reasons for the increased vaccination among students, in particular MS students, were the requirement of an “EU Digital Certificate” and better information about the benefits of vaccination [16]. In the framework of the abovementioned statistics, the 28.4% vaccination rate of AU students constitutes a serious concern.

The available literature with a similar sample composition as in the current study reported rates of 73.3% and 77.81% [16,30]. In our study, 266 (57%) from the MS group demonstrated VA. In previous studies, fields of study related to health science were linked to increased VA [14,15,17,18]. In this study, a significant change related to the field of study was found in the VH group (*p* = 0.0414), indicating that MS students perceived more fear at the beginning of pandemics and increased risk perception in everyday life. This may be related to differences in levels of knowledge related to the study field. The perceived fear and risk, according to [17,31], were not influential factors in vaccination and were related to younger age and the large availability of vaccines at the time the study was performed. Although the results obtained for VA did not show significant change, since students in this group were vaccinated voluntarily, similar to [25,32], who considered fear experienced at the beginning of pandemics and disease severity, we can also consider them as factors contributing to VA.

Observance of protective measures during the pandemic period was among the questions included in this survey. Regardless of whether they were vaccinated voluntarily or mandatorily, the small difference shows that overall, there was a very good attitude toward the observance of protective measures. In this context, our findings are not in line with those of Gallè et al. [32], who found in a study among Albanian undergraduates that more than one-third did not consider either facial masks or the disinfection of surfaces to be effective protective measures. It is expected for students in branches related to medicine to have a more positive attitude towards infection control as they gain more knowledge of the risk factors. The results of our study show similar observance of protective measures among MS and Non-MS students concerning vaccination attitudes, whether voluntary or mandatory. It was found [29] that VH students considered personal protection a substitute for vaccination. In her study among French students, Tavolacci [14] observed that students with negative vaccine intentions were significantly less likely to engage in COVID-19 prevention behaviors of wearing masks and social distancing.

Concerning online lectures, a measure adopted worldwide at the beginning of pandemics, previous studies among Albanian university students by the authors [25,33] reported that online learning was not considered by the majority of students to adequately replace in-class learning and that they were not satisfied. Additionally, the desire to return to classroom teaching was considered “an important driver for increased uptake of vaccine in this population” [27]. If we take into consideration fields of study related to medicine, online learning hinders the possibility of clinical and laboratory practices, which are essential parts of medical education [34]. It is worth remembering that for the entire academic year of 2020–2021, there was only online learning. Contrary to expectations, more than half the MS students in the VA group did not consider that vaccination would enable in-class learning. Interestingly, VH students expressed more conviction that vaccination would enable a return to classroom learning.

At the beginning of the pandemic, misinformation and conspiracy theories focused on the way the virus spread and the protective measures undertaken. As soon as the first vaccines began to be tested, they were surrounded by misinformation and conspiracy theories. In this regard, the result of this study in the VH group is in line with [35]: apart from studying the general adult population or students, it seems to agree on attributing to social media the role of misinformation about preventive attitudes and behaviors and COVID-19 vaccines. Interestingly, the findings regarding VA showed that both groups showed a similar trend of using social networks as a major information source, which seems to be supported by [17], which found a positive association between social media trust and vaccination.

In an Italian study [36] performed among healthcare degree students, aiming to evaluate the impact of vaccination-related extracurricular activity on student knowledge, the authors concluded that this “was highly effective in increasing the students’ knowledge on vaccination: despite good overall scores in the pre-course test, the different groups were able to increase their final score considerably.” According to the same study [36], the next generation of healthcare professionals should receive appropriate education and attain knowledge and technical skills on vaccines and vaccination, as this will help them recommend appropriate vaccinations to patients. Surprisingly, among the VH group, both MS and Non-MS students showed similar levels of knowledge about the protection offered by vaccination, since almost a quarter of them denied the risk posed by non-vaccinated individuals. Similarly, they were not willing to recommend vaccinations to family or friends. On the other hand, among students who vaccinated voluntarily, this study did not find a significant change related to the field of study in knowledge-related protection or in willingness to recommend vaccination [37]. The above findings from our study suggest that this gap in knowledge, especially among MS students, should be addressed by incorporating activities proposed by previous reports as effective in improving attitudes toward vaccination [38].

Students who reported voluntary vaccination had almost the same attitudes: their positive and negative answers were almost divided in half as to the question of whether the decision would have been easier if not mandatory. In other studies [21,39], it was found that there was agreement with the mandatory policy among those willing to receive the vaccine and those who would get vaccinated voluntarily. A total of 46.9% of our sample reported mandatory vaccination. As we considered this group VH respondents, the positive answer to the same question by the majority of both subgroups (89.5% and 89.2%) clearly expresses their disagreement with the mandatory policy.

Hesitation over an unsafe and ineffective vaccine is among the factors contributing to VH [3,5,17,29]. In this study, contrary to [19,30,39], where the most reported safety barrier was found among non-vaccinated students, hesitation over an unsafe and ineffective vaccine was found among voluntarily vaccinated students of both subgroups. Additionally, no significant changes were observed related to the field of study, similar to Gao [36], who found that the percentage of medical students willing to be vaccinated was 54.1%, slightly higher than that of non-medical students.

According to Taye [15], students studying in health science fields were twice as likely to have good willingness to accept the COVID-19 vaccine as students studying non-health subjects. Academic staff involved in teaching MS were attributed, as with healthcare workers, the ability to influence vaccine uptake [5]. Furthermore, according to Kecojevic [27], strong recommendations from healthcare educators and healthcare providers may be critical in promoting vaccine uptake among college students. Hence, expectations for MS students are to show a positive attitude toward vaccination.

The results of this study showed that although VH MS students perceived encouragement from academic staff, they still did not embrace voluntary vaccination [27]. This finding makes us suggest that hesitation over an unsafe and ineffective vaccine prevails over a student’s decision to vaccinate. In an online survey performed among 2328 employees at a multi-campus health sciences university [7], a higher intention from academic staff enabled the author to conclude that medical knowledge and patient care might increase VA. The significant change in the VA group as regards perceived encouragement from academic staff is supported by [5,7,31], attributing medical academic staff a crucial role in increasing VA.

This study has its limitations. As with all similar studies, self-reported data may be subject to reporting bias. Student participation in the study was voluntary, which may introduce selection bias. It was impossible to have an equally distributed sample with regard to sex and study program, which in turn may lead to selection bias. To not bloat the questionnaire, we did not include questions about socioeconomic status or place of residence. This could have generated broader conclusions. In our opinion, another limitation is related to the missing data from our country regarding VH and VA among the general population, which shrinks the conclusions. In addition to its limitations, we strongly believe that this study provides important information regarding factors that affect students’ decisions to vaccinate, especially considering the lack of such information in our country.

## 5. Conclusions

This survey study conducted among Albanian University students provides valuable knowledge about factors related to vaccination against COVID-19.

Vaccine safety and efficacy were hindering factors in vaccination. Additionally, based on the results of this study, the students felt encouraged to vaccinate by the academic staff. This clearly demonstrates that the staff do not lack the skills to enhance students’ knowledge about the risk of infectious diseases and the importance of vaccination. Therefore, to influence students’ attitudes toward vaccination as much as possible, comprehensive educational programs, including modifications of existing curricula, should be considered.

## Figures and Tables

**Table 1 vaccines-11-01215-t001:** General description of the sample, COVID-19-related health experience, and attitudes toward vaccination.

Variables		n (%)	
Age		639 (72.7%)	
18–23		164 (18.6%)	
24–33		164 (18.6%)	
34–43		38 (4.27%)	
44–53		9 (1%)	
**Sex**			
**Female**		**Male**	
**Field of study**			
MS	317 (62.6%)	189 (37.3%)	506 (57.6%)
Non-MS	258 (69.3%)	114 (30.6%)	372 (42.3%)
**COVID-19 health-related experience**			
**Infected in the last 6 months**			
Yes	244 (70%)	103 (29.6%)	347 (39.5%)
No	354 (66.6%)	177 (33.3%)	531 (60.4%)
**Illness experienced in family**			
Yes	203 (68.1%)	95 (31.8%)	298 (33.3%)
No	402 (69.3%)	178 (30%)	580 (66%)
**Attitudes toward vaccination**			
**Regular vaccination in childhood**			
Yes	568 (66.7%)	283 (33.2%)	851 (97%)
No	19 (70%)	8 (29%)	27 (0.03%)
**Vaccinated against COVID-19**			
Yes	514 (66.4%)	259 (33.5%)	773 (88%)
No	74 (70%)	31 (29.5%)	105 (11.8%)
**Voluntary**			
Yes	302 (64.8%)	164 (35.1%)	466 (53%)
No	286 (69.4%)	126 (30%)	412 (46.9%)

**Table 2 vaccines-11-01215-t002:** Comparison between MS * and Non-MS ** students in the VA *** group.

Voluntary Vaccination					
	MS		Non-MS		
Variable	Yes	No	Yes	No	*p* Value
Have you experienced fear at the beginning of pandemics?	199 (74.8%)	67 (22.1%)	146 (73%)	54 (27%)	0.3594
Observance of protective measures	242 (90.9%)	24 (9.1%)	183 (91.5%)	17 (8.5%)	0.1336
Risk perception in everyday life	127 (47.7%)	139 (52.2%)	82 (41%)	118 (59%)	0.056
Fear for one’s life if infected by COVID-19	130 (48.8%)	136 (51.1%)	96 (48%)	104 (52%)	0.8808
Severity of disease in the family	107 (40.2%)	159 (59.7%)	69 (34.5%)	131 (65.5%)	0.2714
Do you think unvaccinated people pose a risk to others?	145 (54.5%)	111 (41.7%)	112 (56%)	88 (44%)	0.818
Online lecture influence	128 (48.1%)	138 (51.8%)	92 (46%)	108 (54%)	0.8336
Influence of friends or classmates	26 (9.7%)	240 (90.2%)	22 (11%)	178 (89%)	0.4778
Would you recommend vaccination?	191 (71.8%)	75 (28.1%)	158 (79%)	42 (21%)	0.1164
Social networks major source of information	201 (75.6%)	65 (24.4%)	149 (74.5%)	51 (25.5%)	0.3174
Easier decision if not mandatory	153 (57.5%)	113 (50.3%)	111 (55.5%)	89 (44.5%)	0.818
Hesitation over unsafe and ineffective vaccine	132 (49.6%)	134 (50.3%)	93 (46.5%)	107 (53.5%)	0.395
Did the academic staff encourage you to vaccinate?	107 (40.2%)	159 (59.7%)	63 (31.5%)	137 (68.5%)	0.034

* Medical science, ** non-medical science, *** vaccine acceptance.

**Table 3 vaccines-11-01215-t003:** Comparison between MS * and Non-MS ** students in the VH *** group.

Mandatory Vaccination					
	MS		Non-MS		
Variable	Yes	No	Yes	No	*p* Value
Have you experienced fear at the beginning of pandemics?	159 (67%)	78 (32.9%)	102 (58.2%)	73 (41.7%)	0.0414
Observance of protective measures	199 (83.9%)	38 (16%)	51 (86.2%)	23 (13.7%)	0.5486
Risk perception in everyday life	102 (43%)	135 (56.9%)	48 (27.4%)	127 (72.5%)	0.002
Fear for one’s life if infected by COVID-19	129 (54.4%)	108 (45.5%)	76 (43.4%)	99 (56.5%)	0.0204
Severity of disease in family	78 (33.3%)	159 (67%)	54 (30.8%)	121 (69.1%)	0.5962
Do you think unvaccinated people pose a risk to others?	45 (18.9%)	192 (81%)	26 (14.8%)	149 (85.1%)	0.2802
Online lecture influence	145 (61.8%)	92 (38.8%)	101 (57.7%)	74 (42.2%)	0.3628
Influence of classmates’ decisions	46 (19.4%)	191 (80.5%)	21 (12%)	154 (88%)	0.7338
Would you recommend vaccination?	47 (19.8%)	190 (80%)	38 (21.%)	137 (78.2%)	0.704
Social networks major source of information	191 (80%)	46 (19.4%)	134 (76.5%)	41 (23.4%)	0.3576
Easier decision if not mandatory	212 (89.5%)	25 (10.5%)	156 (89.2%)	19 (10.8%)	0.7338
Hesitation over unsafe and ineffective vaccine	219 (92.4%)	18 (7.5%)	159 (90.8%)	16 (9.1%)	0.6892
Did the academic staff encourage you to vaccinate?	173 (73%)	64 (27%)	97 (55.4%)	78 (44.5%)	0.002

* Medical science, ** non-medical science, *** vaccine hesitancy.

## Data Availability

Data available on request.

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
