# Peer review of "Determinant Factors of Voluntary or Mandatory Vaccination against COVID-19: A Survey Study among Students at Albanian University"

_vaccines, 2023, doi:10.3390/vaccines11071215_

Round 1
Reviewer 1 Report (Previous Reviewer 3)
The authors have correctly applied the suggestions I gave in the first review.
But looking closely at the tables I realized that they can be improved:
- replace p with p-value
- specify in a note at the end of the table all the acronyms used
- center the numbers in the columns of the tables
- remove the spaces before the round bracket )
- finally I suggest to the authors to put the sex variable of table 1 at the beginning of the column, this will avoid having to repeat the male and female categories several times thus improving the quality of the table
Author Response
Reviewer 1 comments and suggestions
- replace p with p-value
- specify in a note at the end of the table all the acronyms used
- center the numbers in the columns of the tables
- remove the spaces before the round bracket )
- finally I suggest to the authors to put the sex variable of table 1 at the beginning of the column, this will avoid having to repeat the male and female categories several times thus improving the quality of the table
In the revised manuscript we applied all suggestions of reviewer 1.
Reviewer 2 Report (New Reviewer)
In this manuscript titled "Determinant factors of voluntary or mandatory vaccination against Covid-19. A survey study among students at Albanian University" Elona and colleagues aim at comparing determinants of voluntary and mandatory vaccination in students of Albanian University.
Some major issues should be addressed before publication, being the first one the current english form of the manuscript, which is considerably hard to understand. I find very hard to review the article, and I suggest reviewing it carefully. Some sentences even lacks verbs. The same applies for punctuation and form the numbers are written.
I am sure that the manuscript has potential, and it could be resubmitted after a thorough revision.
In certain segments of the manuscript, the level of comprehension in English is considerably challenging. Therefore, I recommend conducting a thorough language review.
Author Response
Following reviewers 2 suggestion the manuscript was sent to English editing service.
Reviewer 3 Report (New Reviewer)
Generally this is an interesting paper and the discussion and conclusion indicate the importance of having medical and other students vaccinated as well as the challenges of achieving this.
Line 21 & also 43,44
“ At the very beginning of the pandemics, the vaccine was considered by health authorities and the med-20ical community the only way to curb the spread of the virus.”
· This is not correct. Along with hand washing, social distancing and wearing of masks, vaccination was considered by medical authorities as a key way to curb the spread of the virus.
NOTE: REFERENCES 23 and 24 do not directly link to the papers with the hyperlink
Please check other references
SEE THE EDITED COMMENTS IN THE ATTACHED .pdf

The English language in this paper does need significant work.
Lines 19 & 42
· This should read: "The world faced serious health and socioeconomic issues with the advent of COVID-19"
Line 34
· the staff do not lack the skills
Line 36
· This should read "to influence students' attitudes as much as possible towards ...
Line 44, 45
Should read
· Efforts to enable the development of vaccines was documented by WHO, counting on April 26 seven candidate vaccines in …
o Note: the paper quoted considers 52 possible vaccine candidates with the seven in the clinical evaluation phase
Line 54
· should read: "disease severity and the crucial role ..."
ENGLISH LANGUAGE NEEDS A SERIOUS EDIT (although it does improve to a certain extent in the discussion section) … these examples above are from the first few paragraphs … I HAVE NOT EDITED LANGUAGE FURTHER (Except in odd spots where I could not help myself)
Author Response
Reviewer 3 comments and suggestions
Line 21 & also 43,44
“ At the very beginning of the pandemics, the vaccine was considered by health authorities and the med-20ical community the only way to curb the spread of the virus.”
- This is not correct. Along with hand washing, social distancing and wearing of masks, vaccination was considered by medical authorities as a key way to curb the spread of the virus.
NOTE: REFERENCES 23 and 24 do not directly link to the papers with the hyperlink
Please check other references
The English language in this paper does need significant work.
Lines 19 & 42
- This should read: "The world faced serious health and socioeconomic issues with the advent of COVID-19"
Line 34
- the staff do not lack the skills
Line 36
- This should read "to influence students' attitudes as much as possible towards ...
Line 44, 45
Should read
- Efforts to enable the development of vaccines was documented by WHO, counting on April 26 seven candidate vaccines in …
o Note: the paper quoted considers 52 possible vaccine candidates with the seven in the clinical evaluation phase
Line 54
- should read: "disease severity and the crucial role ..."
Following reviewers 3 suggestion the manuscript was sent to English editing service.
Regarding the references 23 and 24 that do not directly link to the papers with the hyperlink it happens because we had to modify the link according to references format requested from the journal.
References 23 https://exit.al/en/vaccination-centres-to-be-set-up-at-albanian-universities/
Reference 24 direct link https://exit.al/en/survey-reveals-majority-of-albanian-university-students-against-mandatory-vaccines/
Unfortunately, we could not find other references.
Reviewer 4 Report (New Reviewer)
Concerning the role of the academic staff in encouraging vaccination (Table 2 and 3), the discussion of this result is inadequate and should be supplemented with further explanations based on both previously published data and the author's notion and recommendation. Furthermore, doubts regarding the vaccine's safety and efficacy of the vaccine could be the cause.
Nil
Author Response
Reviewer 4 comments and suggestions
Concerning the role of the academic staff in encouraging vaccination (Table 2 and 3), the discussion of this result is inadequate and should be supplemented with further explanations based on both previously published data and the author's notion and recommendation. Furthermore, doubts regarding the vaccine's safety and efficacy of the vaccine could be the cause.
Following reviewers 4 suggestion in the revised manuscript we added further explanations that are marked by using the “Track Changes” function.
Round 2
Reviewer 2 Report (New Reviewer)
This manuscript by Elona and colleagues is important as it provides valuable insights into the determinants and mandatory vaccination among students at Albanian University. The revised version of the manuscript has improved significantly, with a simple yet effective methodology that is well explained. However, I have a few suggestions to further enhance the manuscript:
- In the methods section, it would be beneficial to specify that the study utilized a convenience sample, highlighting the potential limitations associated with this sampling approach.
- It is advisable to mention that student participation in the study was voluntary, which may introduce selection bias. This clarification will help acknowledge that those who were more inclined towards vaccination might have been more willing to participate.
- Regarding healthcare degree students, it would be valuable to discuss the potential impact of specific educational courses on their knowledge and acceptance of vaccines. Including relevant references (1-2) in this discussion can strengthen the findings and provide additional support to the study's conclusions.
Overall, incorporating these suggestions will further enrich the manuscript and contribute to a more comprehensive understanding of the determinants and implications of mandatory vaccination among students at Albanian University.
1. https://pubmed.ncbi.nlm.nih.gov/35891250/
2. https://pubmed.ncbi.nlm.nih.gov/22761532/
English improved a lot
Author Response
Following reviewer’s comments
- In the methods section, it would be beneficial to specify that the study utilized a convenience sample, highlighting the potential limitations associated with this sampling approach.
- It is advisable to mention that student participation in the study was voluntary, which may introduce selection bias. This clarification will help acknowledge that those who were more inclined towards vaccination might have been more willing to participate.
- Regarding healthcare degree students, it would be valuable to discuss the potential impact of specific educational courses on their knowledge and acceptance of vaccines. Including relevant references (1-2) in this discussion can strengthen the findings and provide additional support to the study's conclusions.
Overall, incorporating these suggestions will further enrich the manuscript and contribute to a more comprehensive understanding of the determinants and implications of mandatory vaccination among students at Albanian University.
- https://pubmed.ncbi.nlm.nih.gov/35891250/
- https://pubmed.ncbi.nlm.nih.gov/22761532/
In the revised version we followed reviewer’s comments by using the “Track Changes” function.
This manuscript is a resubmission of an earlier submission. The following is a list of the peer review reports and author responses from that submission.
Round 1
Reviewer 1 Report
The manuscript entitled "Impact of mandatory policy on the vaccine for Covid-19 uptake. A survey study among students at Albanian University" summarizes an investigation of students of an Albanian university on their attitudes towards COVID-19 vaccination. The methodology of the survey is questionable. and the argumentation included in the manuscript (in all sections) are
Major issues:
1. The aim of the investigation is different in the abstract ("to find out the approach for vaccinations for lives saving and hesitations for these vaccinations through stigma" and in the manuscript text ("to evaluate if the desire to return to present teaching and release of restrictive measures will prevail over factors related to vaccine hesitancy and refusal") are different. And both are vague and unclear!!!
2. The introduction is not relevant. It is not clear how the presented facts on vaccines and vaccine uptake among students refer to the situation in Albania and how it helps justify/introduce the study aim. Only the section in lines 59-71 seem relevant, but is written in a confusing way.
3. The methods are not suffiently described. It is recommended to follow international guidelines for reporting cross-sectional surveys (for example STROBE, https://www.strobe-statement.org/). It is not clear what was the outcome studied, inclusion/exclusion criteria. How the authors checked that those who responded to the survey represent Albanian students or even students of this particular university? Did they attempt to send reminders or study a subset of non-respondents? Anyway, if the aim of the survey is not clear, it is difficult to judge the appropriateness of methods...
4. The results are presented chaotically. It's not clear what is the aim of the study, the main outcome studies, and therefore various descriptive results and tables of associations seem obsolete...
5. The discussion and conclusions are confusing and difficult to follow, like the rest of the manuscript.
Specific comments:
6. The abstract is very unclear. Since it is the most important part that is mostly read, an unclear abstract is really disqualifying any abstract.
7. Introduction, lines 48-51: In one paragraph, authors use three various terms to describe the same indicator: "low rate of vaccine acceptance", "rate of vaccination refusal" and "resistance to vaccination". This is introducing chaos in the argumentation. It is crucial to use terminology in a consistant manner and clearly define at first use.
8. Results, lines 105-106: What does mean "students were vaccinated voluntarily"? It is really confusing in light of the previous information that vaccination was mandatory.
9. Results, lines 111-112: The calculation of the association between mandatory and voluntary vaccination does not make sense, since vaccination can be either voluntary or mandatory... Or I don't understand something?
There are many other problems in this manuscript. Overall, the argumentation is unclear. Since the aim is not clearly defined, it cannot in any way contribute to the international scientific community.
Author Response
Point 1: The aim of the investigation is different in the abstract ("to find out the approach for vaccinations for lives saving and hesitations for these vaccinations through stigma" and in the manuscript text ("to evaluate if the desire to return to present teaching and release of restrictive measures will prevail over factors related to vaccine hesitancy and refusal") are different. And both are vague and unclear!!!
Response 1: In the revised manuscript the aim in abstract and in the manuscript text are similar: “to compare determinants of voluntary and mandatory vaccination among students of Albanian University”.
Point 2: The introduction is not relevant. It is not clear how the presented facts on vaccines and vaccine uptake among students refer to the situation in Albania and how it helps justify/introduce the study aim. Only the section in lines 59-71 seem relevant, but is written in a confusing way.
Response 2: Data regarding the situation in Albania are almost missing in the literature. Also, since there was a mandatory policy but in the same time there was also students that vaccinated voluntary we aim to compare if they had different attitudes. With regard to situation in Albania, in the revised manuscript we included few data available in the international literature.
Point 3: The methods are not suffiently described. It is recommended to follow international guidelines for reporting cross-sectional surveys (for example STROBE, https://www.strobe-statement.org/). It is not clear what was the outcome studied, inclusion/exclusion criteria. How the authors checked that those who responded to the survey represent Albanian students or even students of this particular university? Did they attempt to send reminders or study a subset of non-respondents? Anyway, if the aim of the survey is not clear, it is difficult to judge the appropriateness of methods..
Response 3: The inclusion criteria was to be enrolled in Albanian University for the academic year 2021-2022. As the questionaire was sent to students from AU mailing list there was no need to further check. 3 reminders were sent during the week the form was active.
Point 4: The results are presented chaotically. It's not clear what is the aim of the study, the main outcome studies, and therefore various descriptive results and tables of associations seem obsolete..
Response 4: In the revised manuscript the results are presented according to voluntary (table 2) and mandatory (table 3) vaccination.
Point 5: The discussion and conclusions are confusing and difficult to follow, like the rest of the manuscript.
Response 5: After revision discussion ane conclusion are in accordance with the study aim.
Point 6: The abstract is very unclear. Since it is the most important part that is mostly read, an unclear abstract is really disqualifying any abstract.
Response 6: Following reviewers recommandation to extensively revise the manuscript the abstract is revised too.
Point 7: Introduction, lines 48-51: In one paragraph, authors use three various terms to describe the same indicator: "low rate of vaccine acceptance", "rate of vaccination refusal" and "resistance to vaccination". This is introducing chaos in the argumentation. It is crucial to use terminology in a consistant manner and clearly define at first use
Response 7: In the revised manuscript we used same terminology “vaccine hesitancy”
Point 8: Results, lines 105-106: What does mean "students were vaccinated voluntarily"? It is really confusing in light of the previous information that vaccination was mandatory.
Response 8: Although there was a mandatory policy there were also students that vaccinated voluntary.
Point 9: Results, lines 111-112: The calculation of the association between mandatory and voluntary vaccination does not make sense, since vaccination can be either voluntary or mandatory... Or I don't understand something?
Response 9: In the revised manuscript a different statistical method was used.
Reviewer 2 Report
This is an interesting study that sheds light on the situation among selected Albanian students. To upgrade the paper, I suggest to:
1. Provide more information on the methodology and tools, and statistical tests used (which questionnaire was used? It is said that it is based on literature- more information is needed).
2. Table 1- provide the data in a way that reveals the representation of the target population (how many males, females, etc.).
3. Re-think what is the study's conclusion: is it that anti-vexers tend to vaccinate less? What is new about it?
4. Was there any difference in compliance based on the types of vaccines?
5. Use compliance rates, refusal, etc.
Author Response
Point 1: Provide more information on the methodology and tools, and statistical tests used (which questionnaire was used? It is said that it is based on literature- more information is needed)
Response 1: In the revised version references regarding the literature used for the questionnaire are included
Point 2: Table 1- provide the data in a way that reveals the representation of the target population (how many males, females, etc.).
Response 2: In the revised manuscript in table 1 data include distribution detailed distribution among males and females as well as the field of study.
Point 3: Re-think what is the study's conclusion: is it that anti-vexers tend to vaccinate less? What is new about it?
Response 3: In the revised manuscript we compared attitudes of students that vaccinated voluntary and student who vaccinated due to the mandatory policy.
Point 4: Was there any difference in compliance based on the types of vaccines?
Response 4: No reference to any specific vaccine was included in the questionnaire
Point 5:. Use compliance rates, refusal, etc.
Response 5: After the revisions we expressed the result as vaccine acceptance and hesitancy
Reviewer 3 Report
Unfortunately, the manuscript has several methodological and statistical lacks, so it is not suitable for publication in Vaccines.
The manuscript does not show whether the project was submitted to the opinion of the Ethics Committee as required by law
The aim is expressed differently in various parts of the manuscript so it is confusing.
The authors do not explain in the manuscript how they validated the questionnaire
The word gender is erroneously used by the authors as if it were synonymous with the word sex
The verb tenses in the results paragraph are sometimes used in the present and sometimes in the past
The results are reported using several digits after the decimal point that don't make sense
The authors applied the chi-square test to investigate the relationship between qualitative variables, but did not estimate the strength of the association, for example through Cramer's V
The authors erroneously report the p-value relating to the accepted alpha error level (p<0.05) together with the p-value related to the tests used for the statistical analysis. Also, the p-value is erroneously written with a capital "p".
Author Response
Point 1: The manuscript does not show whether the project was submitted to the opinion of the Ethics Committee as required by law
Response 1: In the revised manuscript aproval from Ethics Committee is included
Point 2: The aim is expressed differently in various parts of the manuscript so it is confusing
Response 2: After extensive revision among corrections you will find that the aim is corrected
Point :3 The authors do not explain in the manuscript how they validated the questionnaire
Response 3: The questionnaire was validated from some of the authors of the study and the statistician.
Point 4: The word gender is erroneously used by the authors as if it were synonymous with the word sex
Response 4: We provided to correct
Point 5: The verb tenses in the results paragraph are sometimes used in the present and sometimes in the past
Response 5: In the revised manuscript we have correct this point
Point 6: The results are reported using several digits after the decimal point that don't make sense
Response 6: After revisions this point is also corrected
Point 7: The authors applied the chi-square test to investigate the relationship between qualitative variables, but did not estimate the strength of the association, for example through Cramer's V
Response 7: Following reviewers recommandation to extensively revise the manuscript the statistical method was changed.
Point 8: The authors erroneously report the p-value relating to the accepted alpha error level (p<0.05) together with the p-value related to the tests used for the statistical analysis. Also, the p-value is erroneously written with a capital "p".
Response 8: p-value is written correctly in the revised manuscript